# Dextranase Production Using Marine *Microbacterium* sp. XD05 and Its Application

**DOI:** 10.3390/md21100528

**Published:** 2023-10-07

**Authors:** Hind Boualis, Xudong Wu, Boyan Wang, Qiang Li, Mingwang Liu, Lei Zhang, Mingsheng Lyu, Shujun Wang

**Affiliations:** 1Jiangsu Key Laboratory of Marine Bioresources and Environment/Jiangsu Key Laboratory of Marine Biotechnology, Jiangsu Ocean University, Lianyungang 222005, China; hindboualis@gmail.com (H.B.); xudwu@jou.edu.cn (X.W.); boyanwang@jou.edu.cn (B.W.); qiangli@jou.edu.cn (Q.L.); mwliu@jou.edu.cn (M.L.); mslyu@jou.edu.cn (M.L.); 2Co-Innovation Center of Jiangsu Marine Bio-Industry Technology, Jiangsu Ocean University, Lianyungang 222005, China

**Keywords:** dextranase, *Microbacterium* sp., enzymatic properties, oligo-maltosaccharines, corn porous starch

## Abstract

Dextranase, also known as glucanase, is a hydrolase enzyme that cleaves α-1,6 glycosidic bonds. In this study, a dextranase-producing strain was isolated from water samples of the Qingdao Sea and identified as *Microbacterium* sp. This strain was further evaluated for growth conditions, enzyme-producing conditions, enzymatic properties, and hydrolysates. Yeast extract and sodium chloride were found to be the most suitable carbon and nitrogen sources for strain growth, while sucrose and ammonium sodium were found to be suitable carbon and nitrogen sources for fermentation. The optimal pH was 7.5, with a culture temperature of 40 °C and a culture time of 48 h. Dextranase produced by strain XD05 showed good thermal stability at 40 °C by retaining more than 70% relative enzyme activity. The pH stability of the enzyme was better under a weak alkaline condition (pH 6.0–8.0). The addition of NH_4_^+^ increased dextranase activity, while Co^2+^ and Mn^2+^ had slight inhibitory effects on dextranase activity. In addition, high-performance liquid chromatography showed that dextran is mainly hydrolyzed to maltoheptanose, maltohexanose, maltopentose, and maltootriose. Moreover, it can form corn porous starch. Dextranase can be used in various fields, such as food, medicine, chemical industry, cosmetics, and agriculture.

## 1. Introduction

Dextrans are complex polysaccharides composed of long chains of glucose molecules linked together [1]. Dextran is a renewable and biodegradable chemical resource that is non-toxic, stable, and water-soluble [2]. Dextranases, also known as α-1,6-D-glucan-6-glucanohydrolase or alpha-glucosidase, are proteases that can hydrolyze the dextrans by breaking down the α-1,6 glycosidic bond and forming oligosaccharides [3]. Dextranases can be produced by mold, yeast, and bacteria [4]. In particular, marine bacteria can produce enzymes with distinctive properties which are not found in enzymes produced by their terrestrial counterparts [5,6]. Research has proved that dextranases from different sources can be radically different. For instance, fungal dextranases have a high level of stability and activity; however, the fermentation process may result in the generation of dangerous byproducts and can be relatively time-consuming [7]. Marine bacterial dextranases are known to be resistant to alkaline environments and tolerant to high salt concentrations [8]. They can also exhibit efficient catalytic activity at moderate-to-high temperatures [9]. Bacteria-derived dextranases generally exhibit lower activity and stability compared to those derived from fungi [7,10]. However, the fermentation process for bacterial dextranases is usually shorter and results in fewer harmful byproducts [11]. However, the effectiveness of dextranases can be influenced by various factors, such as the level of acidity, the content of solids, temperature, contact time, agitation, dextran concentration, as well as the source, activity, and dosage of the enzyme [12,13]. This research aimed to explore the synthesis of dextranase through marine bacteria. Promising marine bacterial strains were isolated, and their morphological and biochemical characteristics were analyzed. Furthermore, optimal growth conditions and production properties of this strain were investigated, followed by experiments to evaluate the effectiveness of marine bacterial dextranase on porous starch. This research provides insights into the potential of marine bacteria as a source of dextranase and the application of marine bacterial dextranases in food and biotechnology industries.

## 2. Results

### 2.1. Screened Dextranase-Producing Strains

The hydrolytic abilities of six marine strains were compared using the blue dextran plate method (Figure 1), and strain XD05, which exhibited the highest dextranase-producing ability, was selected.

### 2.2. Identification of the XD05 Strain

Table 1 shows the morphological and biochemical traits of the XD05 strain. This strain was observed to be a short rod-shaped, Gram-negative, aerobic organism that grew well at 37 °C. The 16S rDNA gene sequence of XD05 was obtained, and it was observed to have a length of 1800 bp (Figure 2a). This sequence was then analyzed by BLAST and compared to existing sequences in the Genbank database. A phylogenetic tree (Figure 2b) was constructed using the MEGA system by aligning the 16S rDNA sequences of the XD05 strain with other existing sequences in the database. The phylogenetic tree revealed that the XD05 strain shared its highest homology with *Microbacterium* sp. B015-10.

### 2.3. Culture Conditions of Strain XD05

#### 2.3.1. Effects of Carbon and Nitrogen Sources on the Growth of the Strain

The effects of different carbon sources and nitrogen sources on dextranase-producing strain XD05 are shown in Figure 3. Yeast extract was found to be the best carbon source for the growth of strain XD05, while soluble starch and corn starch had a minimal effect on the dextranase-producing strain. On the other hand, sucrose, glucose, and lactose promoted the growth of strain XD05 to some extent. As for the nitrogen source, sodium nitrate was found to be the most beneficial nitrogen source for the growth of the strain. Urea, ammonium sulfate, ammonium chloride, tryptone, and peanut also considerably promoted the growth of strain XD05.

#### 2.3.2. Effects of Temperature, pH and Other Parameters on the Growth of the Strain

The optimum growth temperature was 35 °C (Figure 4a). The strain XD05 could grow at a wide pH range, but a pH of 6.0 was optimum (Figure 4b). The strain was grown in a medium containing 2–6% NaCl at an optimum concentration of 3% and did not grow without NaCl (Figure 4c). The strain grew exponentially between 3 h and 18 h (Figure 4d), and then the growth rate stabilized with a slight drop after 48 h.

### 2.4. Optimization of Dextranase Production by Strain XD05

#### 2.4.1. Effects of Carbon and Nitrogen Source on Dextranase Production

Figure 5 shows the impact of various carbon and nitrogen sources on dextranase production using strain XD05. Among the carbon sources, sucrose was found to be the most effective for dextranase production, while maltose had a negligible effect on dextranase production. On the other hand, taro starch, yeast extract, and glucose also promoted dextranase production. Among these nitrogen sources, ammonium sodium was found to be the most favorable source for dextranase production, followed by proteose peptone and soybean peptone. On the other hand, beef broth was deemed unsuitable for large-scale fermentation and dextranase production. Hence, sucrose and ammonium sulfate were selected as the carbon and nitrogen sources, respectively, for optimal dextranase production.

#### 2.4.2. Effects of Temperature and pH on Dextranase Production

The optimal temperature for producing dextranase using strain XD05 was 40 °C. When the temperature was higher or lower, the dextranase-producing capacity of the strain changed considerably (Figure 6a). To determine the optimal pH for dextranase production, the initial pH of the medium varied in the range of 6.0–10.0, and dextranase activity was measured after 48 h of fermentation at different pHs. The highest dextranase activity was observed at an optimum pH of 7.5 (Figure 6b). 

### 2.5. Characterization of Dextranase Activity

#### 2.5.1. Effects of Temperature and pH on Dextranase Activity and Stability

The effect of temperature on dextranase activity is shown in (Figure 7c). Dextranase activity was observed to be very sensitive to fluctuations in temperature, and the optimum temperature was found to be 40 °C. Dextranase maintained a high activity level in the pH range of 6.0–8.0. However, the highest dextranase activity was observed at a pH of 7.0. In addition, the dextranase was very stable in the pH range of 4.0–8.0 (Figure 7b). The thermal stability experiment showed that the residual activity of dextranase remained at almost 80%, following storage at 40 °C (pH 8.0) for 5 h. However, nearly 50% of dextranase activity was lost following storage at 45 °C for 5 h (Figure 7d).

#### 2.5.2. Effects of Metal Ions on Dextranase Activity

The effects of various metal ions on dextranase activity are summarized in Table 2. After the addition of NH_4_^+^, dextranase activity increased from 76.18% to 116.77%. Mg^2+^ and K^+^ also significantly promoted dextranase activity. However, metal ions Co^2+^ and Mn^2+^ showed slight inhibitory effects on dextranase activity. Moreover, Ca^2+^ and Ba^2+^ did not have any significant effect on dextranase activity. 

#### 2.5.3. Substrate Specificity and Final Hydrolysis Products

Figure 8 displays the results of high-performance liquid chromatography (HPLC), revealing the products generated at various time points during the hydrolysis of dextran T2000 using dextranase. The primary hydrolysates detected were maltoheptaose, maltohexaose, maltopentose and maltotriose. As shown in Table 3, the peak areas of these hydrolysates were quantified using HPLC. The peak of maltoheptaose was observed between 0.5 and 4 h, exhibiting endo-dextranase activities. 

### 2.6. Effect of Dextranase on Starch Porosity

The scanning electron microscope (SEM) images of the cavities created by the enzyme within the starch molecules can be seen in Figure 9. The hydrolysis of pure corn starch via dextranase increased significantly with the increase in reaction time. When pure starch was subjected to an 8 h reaction with dextranase, some of the pure starch was over-hydrolyzed, leading to partially broken starch granules. This could be due to the fact that dextranase penetrated the internal structure of each starch molecule and hydrolyzed its amorphous and incomplete crystal regions more effectively. As the enzyme activity increased with time, the growth rate of the inner and surface holes increased, causing the pores to expand further and making the broken starch particles more visible. Due to the action of dextranase, the surface of starch became porous. The water and oil absorption ratee of corn porous starch are given in Table 4. As the enzymatic hydrolysis time continued, the water and oil absorption rate of starch also fluctuated. This may be because the probability of water and oil molecules entering the interior of the starch increased due to the pores, which further enhanced the water and oil absorption by starch. 

## 3. Discussion

*Microbacterium* sp. is newly found marine bacteria with little known about its features. The strain XD05 did not grow in the absence of NaCl, indicating that this strain is a halophilic bacterium, which can grow better at a pH range of 6.0–8.0. NH_4_^+^ enhanced the dextranase activity, which was dissimilar to the dextranase activity pattern shown by *Catenovulum agarivorans* MNH15 in a previous study [14], whereas Co^2+^, Cu^2+^, and Li^+^ exerted opposite effects on dextranase activity. As established by previous research, marine bacteria are generally adapted to low-nutrient conditions and have a relatively lower requirement of medium components for enzyme production. This makes them valuable for industrial applications. *Microbacterium* sp. XD05 demonstrated strong potential for its use in industrial settings due to its unique properties and suitability for enzyme production. Alkaline conditions (pH = 7.5) were found for the dextranase activity of strain XD05, which is similar to the previously reported optimum pH of 8.5 for *Microbacterium* sp. [15]. Similarly, a study reported that an alkaline condition with pH = 12 was favorable for *Microbacterium* sp. *SKS10* [16]. The dextranase produced by strain XD05 was incubated at 40 °C in a water bath for 5 h, and the enzyme was able to retain 80% of its relative enzyme activity. These stability traits are similar to the characteristics of dextranase produced by *Chaetomium globosum* [17]. The products of XD05-dextranase-catalyzed starch hydrolysis included maltoheptaose, malthexaose, maltpentose, and maltotriose. The main hydrolysate produced by the action of XD05 dextranase were isomalto-oligosaccharides, which have a significant promoting effect on the growth of probiotics [18] and the delay of gastric emptying, resulting in long satiety periods and the retardation of the nutrient absorption rate in the small intestine [19,20]. These compounds possess the ability to stimulate the selective growth of probiotic-like bacteria that are normally present in the gut, resulting in improved host health [21,22]. 

Porous starch is a starch modification product that consists of sufficient pores and channels that are distributed on the starch molecules and extend toward the central cavities, which have no impact on the granular structure of starch [23]. There are many ways to prepare porous starch; however, enzymatic modification is a suitable method to control the hydrolysis process, the generation of by-products and hydrolysates, and the product yields [24]. These pores provide space to accommodate special molecules, such as volatile and flavored essential oils, within the starch granules [23,25]. Compared to native starch, porous starch exhibits a better adsorption efficiency, solubility, and swelling capacity [26]. Therefore, porous starch can be used to adsorb, encapsulate, and release different substances [24,27]. Since porous starch is non-toxic, fully biodegradable and environmentally friendly, it can be widely used as an adsorbent in food, medicine, chemicals, cosmetics, agriculture and other related industries [28]. The demands for enzymatically digested starch are increasing in parallel to the rapid development of these industries. The action of dextranase produced by *Microbacterium* sp. XD05 resulted in a hydrolyzed surface of pure corn starch with a porous structure. The adsorption of oil and water by porous starch is considered physical adsorption without selectivity. The adsorption properties of porous starch in this study significantly improved after the enzyme treatment. The water absorption capacity of porous starch reached a maximum value of 100.02 ± 0.08% after 6 h, while oil adsorption capacity reached 100 ± 1.06% after 12 h. This excellent adsorption capacity can be used to enhance the dissolution rate of poorly soluble drugs. Alternatively, products produced via dextranase can be used as a shell material to improve the stability and maintain the amorphous state of drugs [29,30], improve drug wettability, and avoid the aggregation of drug particles [31]. 

## 4. Materials and Methods

### 4.1. Materials

#### Samples and Chemicals

Samples of sea mud, seaweed, and seawater were gathered from the Yellow Sea located in Qingdao, China. Blue Dextran 2000 was obtained from GE Healthcare, and the bacterial genome extraction kit was sourced from Tiangen Biochemical Technology (Beijing, China). The reagents used for physiological and biochemical characterization were acquired from Beijing Luqiao Technology Co., Ltd. (Beijing, China). Additional chemical reagents were obtained from Sinopharm Chemical Reagent Corp (Shanghai, China).

### 4.2. Methods

#### 4.2.1. Screening of Dextranase-Producing Marine Bacterial Strains

To identify marine bacteria with the ability to produce dextranase, a blue dextran plate was used. The screening medium was composed of 10 g of dextran, 1 g of yeast extract, 5 g of peptone, 2 g of blue dextran 2000, 20 g of agar powder, and 1 L of aged seawater with an initial pH of 8.0. The sea mud, seaweed, and seawater suspension were diluted using sterile distilled water and spread evenly on the blue dextran plate. The plate was then incubated at 37 °C for 2 days. The dextranase-producing strain was identified by observing the transparent zones around the colony. The diameters of the colony and transparent zone were measured and compared to screen the strain. The selected strains were then cultured in a fermentation shake flask (in a culture medium, excluding agar and blue dextran 2000). After 48 h of fermentation at 25 °C and 180 rpm, dextranase activity was measured using an adapted method [12]. The strain with the highest dextranase activity was selected for further study. 

#### 4.2.2. Identification and Characterization of the Bacteria

The morphological characteristics of strain XD05 were determined by examining the colonies of the strain using light microscopy and SEM. The genome of strain XD05 was extracted using a bacterial genome extraction kit, and the 16S rDNA gene was amplified using the polymerase chain reaction (PCR). The extracted plasmids and PCR products were verified using agarose gel and sent for sequencing (Sangon, Shanghai, China). The obtained 16S rDNA gene sequence was compared to existing sequences in GenBank using BLAST analysis. Phylogenetic trees were constructed using the neighbor-joining method to identify the strain.

#### 4.2.3. Culture Conditions for the Growth of the Strain

For the growth of this strain, a culture broth was prepared by mixing 5 g of yeast extract, 5 g of peptone, and 10 g of dextran in 1 L of aged seawater (pH of 8.0). The activated strain was then cultured in this broth at 30 °C and 180 rpm for 48 h. The resulting supernatant was collected to measure dextranase activity.

#### 4.2.4. Culture Conditions for Dextranase Production and Purification

For dextranase production, the medium containing the yeast extract, peptone, dextran, and seawater was used (pH 8.0). After the activation of this strain, 2% of inoculum was introduced into the dextranase-producing medium, and this medium was incubated at 35 °C and 180 rpm for a duration of 48 h. Following these steps, the fermentation of the broth was purified. Initially, it was centrifuged at 10,000 rpm for 30 min at 4 °C. The supernatant was then filtered through a 0.45 µm membrane and subjected to further purification via ultrafiltration using a hollow fiber column with a molecular cutoff weight of 30 kDa. Finally, the resulting supernatant was analyzed to determine dextranase activity.

#### 4.2.5. Enzyme Assay

The dextranase enzyme solution of 50 μL was combined with 75 μL of 3% dextran T20 in a 75 mM pH 6.0 Tris-HCl buffer, and this mixture was incubated at 40 °C for 30 min. Subsequently, the concentration of reduced sugar in the mixture was determined using the DNS method [12]. The enzyme activity was defined as the quantity of the enzyme needed to catalyze the release of 1 μmol of maltose per minute under the above-mentioned reaction conditions.

#### 4.2.6. Effects of Carbon and Nitrogen Sources on Dextranase Production 

To determine suitable carbon and nitrogen sources for dextranase production, the yeast extract and peptone present in the fermentation medium were substituted with 10 g/L of various carbon sources, such as corn starch, dextran, glucose, lactose, soluble starch, sucrose, maltose, and yeast extract with 5 g/L of different nitrogen sources like soy peptone, beef extract, sodium nitrate, soybean, tryptone, peanut, ammonium sulfate, ammonium chloride, urea, altering one source at a time. After incubation at 35 °C and 180 rpm for 48 h, the activity of produced dextranase was measured as described in Section 

#### 4.2.7. Effects of Initial pH, and Temperature on Dextranase Production

With all other variables held constant, 2% of the strain inoculum was cultivated under varying temperatures (20–45 °C) for 48 h. To determine the optimal initial pH for dextranase production, the activated strain was cultured for 48 h at different initial pH values (5.0–9.0) and in the medium described in Section 4.2.3. After 48 h of incubation, the activity of produced dextranase was measured.

#### 4.2.8. Effects of Temperature on Dextranase Production Activity and Stability

To determine the optimal temperature range for dextranase activity, enzymatic activity was measured at temperatures of 20 °C, 25 °C, 30 °C, 35 °C, 40 °C, and 45 °C. To assess the thermal stability of dextranase, the enzyme solution was kept at 25 °C, 30 °C, 35 °C, 40 °C, 45 °C and 50 °C for 5 h. Samples were collected every 1 h and stored at 4 °C. Finally, the dextranase enzyme activities of all samples were measured according to the method described before.

#### 4.2.9. Effects of pH on Dextranase Activity and Stability

To determine the optimal pH range for dextranase activity, enzymatic activity was measured at various pH levels (Tris-HCl, pH 5.0–9.0) at an optimum temperature. The stability of pH for the dextranase was also evaluated by preheating the enzyme in different pH buffers (acetate buffer, pH 4.0–6.0; Tris-HCl, pH 6.0–9.0) at a suitable temperature for 4 h in the water bath. The samples were collected and stored at 4 °C, and the dextranase activity of all samples was tested according to the method described earlier. 

#### 4.2.10. Analysis of Final Hydrolysis Products

HPLC was employed to identify the hydrolysates produced at different time points during the hydrolysis of dextran using dextranase. To prepare the standard sugar solutions, 20 mg of standard sugars (G1–G7) were weighed and dissolved in 1 mL of deionized water using a vortex instrument. Gradient dilutions of different standard sugar solutions were then prepared, resulting in samples with sugar concentrations ranging from 20 mg/mL to 0.1 mg/mL. The samples were filtered using a disposable injection needle and a 0.22 μm filter membrane, followed by ultrasonic treatment for 30 min. The enzyme solution was mixed with dextran in a volume ratio of 1:2 and added to a centrifuge tube. The solution was left at 100 °C to allow dextran to hydrolysis at various time intervals (30 min, 1 h, 2 h, 3 h, and 4 h). After the completion of the reaction, the mixture was placed in a boiling water bath for 5 min, filtered through a 0.22 μm filter membrane, and stored at 4 °C for further experimentation. All samples were subjected to HPLC analysis using a waste sugar-park1 (6.5 × 300 mm) chromatographic column. Water was used as the mobile phase. The flow rate was 0.4 mL/min, the column temperature was 75 °C, and the injection volume was 20 μL. The standard compounds used for quantification were glucose, maltose, maltotriose, maltotetraose, maltopentose, maltohexaose and maltoheptaose. The peak areas were measured to quantify each substance. Empower GPC software (Waters, Milford, MA, USA) was used for data acquisition and processing.

### 4.3. Preparation of Porous Starch

To prepare the starch suspension, 10 g of dry starch was mixed with 40 mL of a sodium acetate buffer (0.1 M, pH 6:0) and subjected to incubation at 40 °C for a duration of 30 min, in a water bath equipped with a thermostatic oscillator. Subsequently, dextranase produced by this strain was introduced into the starch suspension, followed by incubation in the water bath for 6, 12, 24, 36, and 48 h at 40 °C. To stop the enzyme’s activity, 6 mL of 95% ethanol was added to reaction solutions and then centrifuged at 5000 rpm/min for 10 min. The resulting sediment, which contained starch, was dried for 48 h at 40 °C, grounded, and placed in a desiccator for further analysis.

### 4.4. Water and Oil Absorption

In total, 0.5 g of starch sample was placed in a dry centrifuge tube (10 mL) and weighed as *W*2. Subsequently, 7 mL of water or oil was added to the tube and vortexed for 5 min at room temperature. Then, the tubes containing starch and oil/water were left at room temperature for 30 min to allow oil or water adsorption by starch. The mixture was then centrifuged at 5000 rmp/min for 15 min, and the weight of the precipitate was measured as *W*1. The water or oil absorption (AB) capacity of starch was calculated using the following equation: AB = 𝑊2 − 𝑊1/𝑊1 × 100%(1)

## 5. Conclusions

In conclusion, the XD05 strain of marine *Microbacterium* sp. was isolated from samples collected from the Qingdao Sea, China, and screened for dextranase production ability. The dextranase showed maximum activity at 40 °C and an initial pH of 7.5. The purified enzyme was stable over a wide range of pH (6.0–8.0) and temperature conditions. Additionally, the enzyme was found to be NaCl-dependent and exhibited substrate specificity toward dextran. Overall, this study provided valuable insights into the enzymatic properties of dextranase produced by a marine bacterium. The dextranase produced by these bacteria has potential applications in various industries, such as food, pharmaceutical, and biotechnology. However, further research is required to optimize the enzyme production and purification process to enhance its catalytic efficiency and stability for industrial applications.

## Figures and Tables

**Figure 1 marinedrugs-21-00528-f001:**
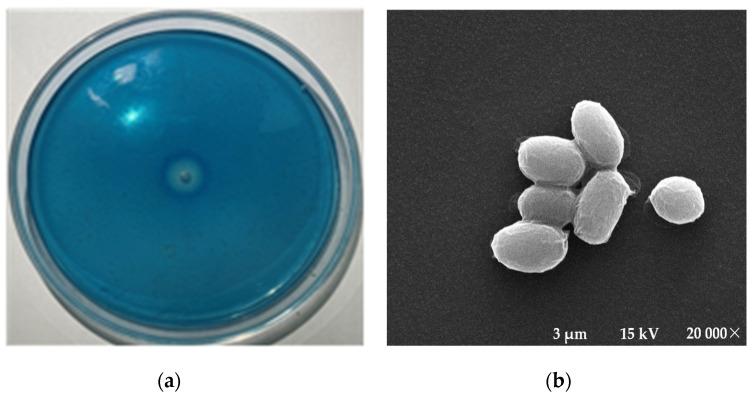
Morphology of the XD05 strain: (**a**) XD05 on a blue dextran plate; (**b**) Scanning electron micrograph.

**Figure 2 marinedrugs-21-00528-f002:**
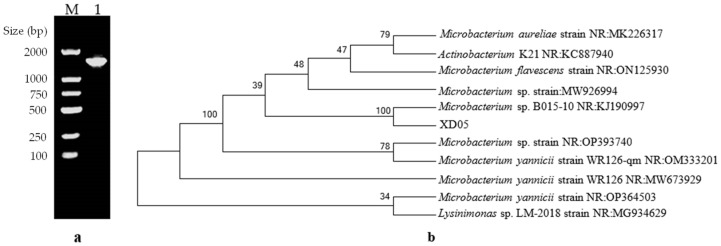
(**a**) Agarose electrophoresis profiles of 16S rDNA PCR products; M: DNA marker; 1: strain XD5 16S rDNA PCR amplification product; (**b**) Phylogenetic tree based on 16S rDNA gene sequences.

**Figure 3 marinedrugs-21-00528-f003:**
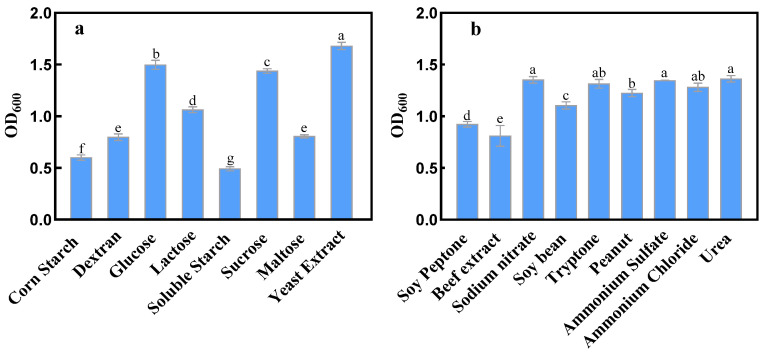
Effect of carbon (**a**) and nitrogen (**b**) sources on the growth of strain XD05 after 24 h. Differing letters within each subfigure (i.e., a, b, c) indicate statistically significant differences (*p* < 0.01). The statistical analysis was performed based on the experimental data of 3 repeats.

**Figure 4 marinedrugs-21-00528-f004:**
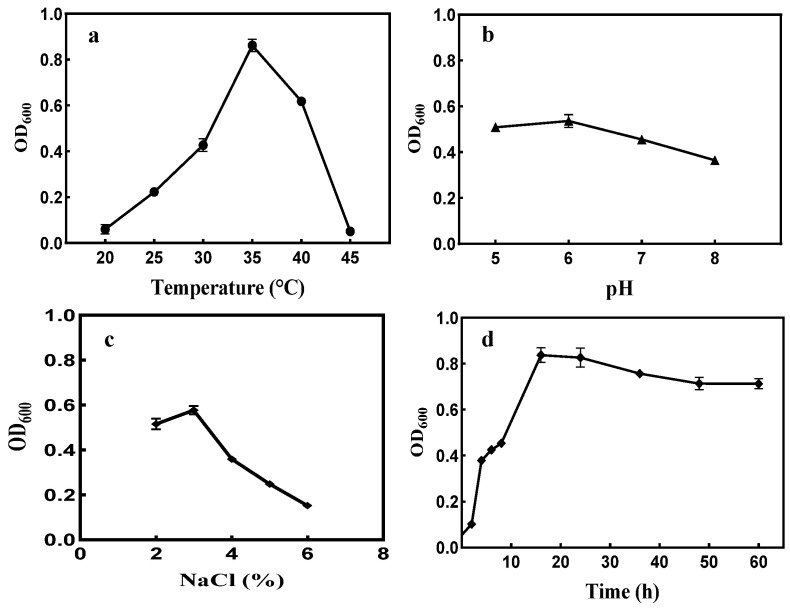
Effect of temperature (**a**), pH (**b**), NaCl concentration (**c**), and time (**d**) on the growth of the strain XD05.

**Figure 5 marinedrugs-21-00528-f005:**
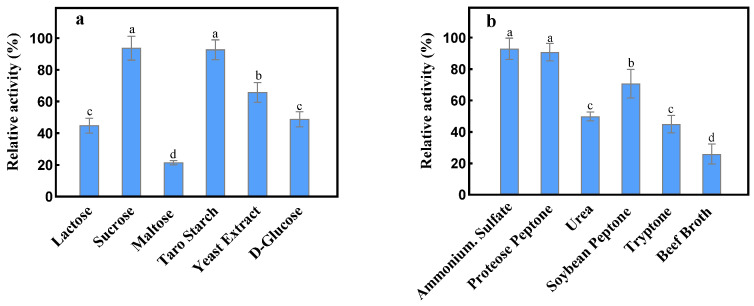
Effect of (**a**) Carbon and (**b**) Nitrogen sources on dextranase production using strain XD05. Differing letters within each subfigure (i.e., a, b, c) indicate statistically significant differences (*p* < 0.01). The statistical analysis was performed based on the experimental data of 3 repeats.

**Figure 6 marinedrugs-21-00528-f006:**
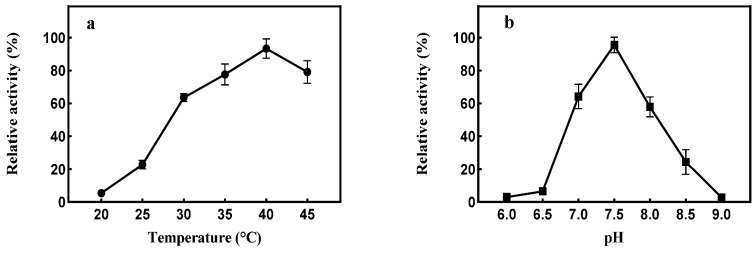
Effects of temperature (**a**) and initial pH (**b**) on dextranase production.

**Figure 7 marinedrugs-21-00528-f007:**
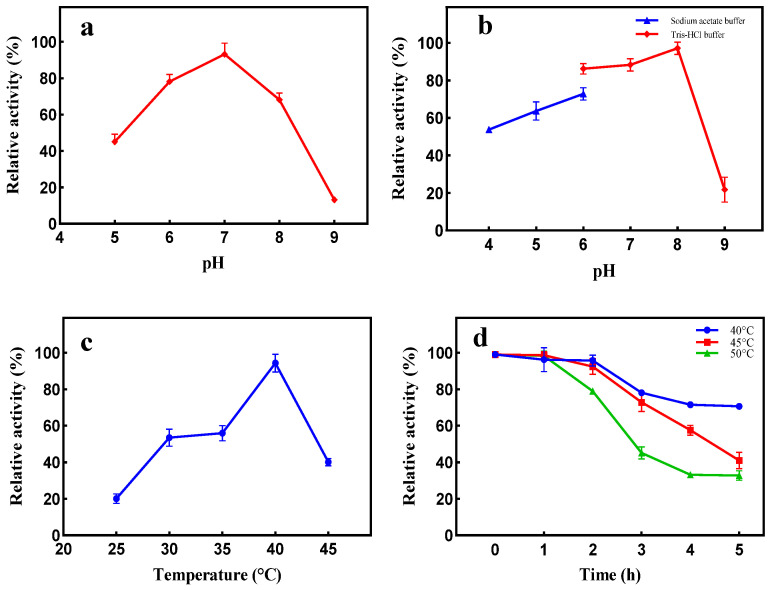
Effects of pH (**a**) and pH and buffer (**b**) on dextranase activity; effects of temperature (**c**) and storage time and temperature (**d**) on dextranase stability.

**Figure 8 marinedrugs-21-00528-f008:**
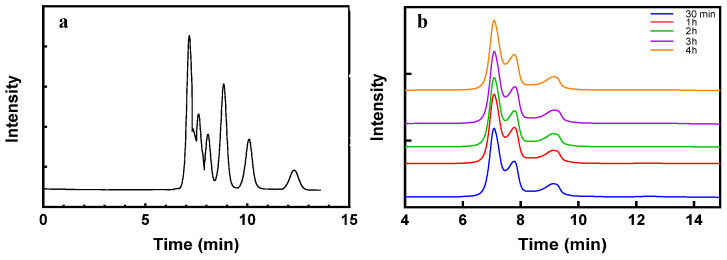
(**a**) The chromatogram of the sugar standards and (**b**) The chromatography of the dextranase hydrolysis products.

**Figure 9 marinedrugs-21-00528-f009:**
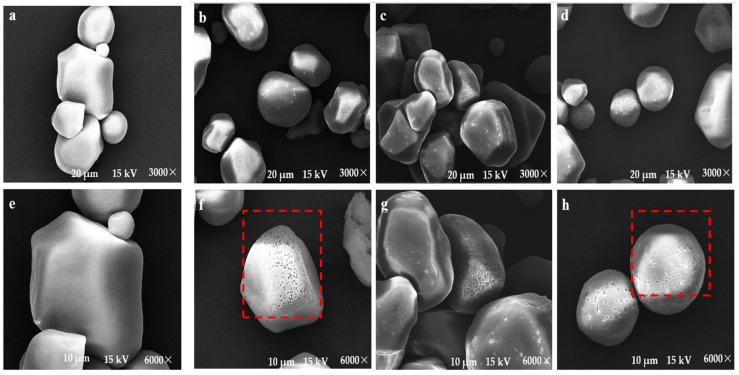
Scanning electron microscopy of pure starch before (**a**,**e**) and after (**b**–**d**,**f**–**h**) enzymatic digestion with XD05 dextranase for 48 h. ((**a**–**d**) 3000× magnification; (**e**–**h**) 6000× magnification).

**Table 1 marinedrugs-21-00528-t001:** Physiological and biochemical characteristics of the strain XD05.

Item	Result	Item	Result
Colony color Shape 4 °C	Milky Rod −	Phenylalanine Mannose Lactose	− + +
37 °C	+	Sorbitol	−
Arginine dihydrolase	−	Sucrose	+
Laetrile broth	+	Glucose	+
Maltose	+	Arabic gum	+
Nitrate (Reduction)	−	Inositol	+

Note: +: Positive; −: Negative.

**Table 2 marinedrugs-21-00528-t002:** Effect of metal ions on dextranase.

	Relative Activity (%) (1 mM)	Relative Activity (%) (10 mM)
Control	100 ± 0.20	100 ± 0.20
Ca^2+^	114.13 ± 1.32	115.08 ± 1.87
NH_4_^+^	76.18 ± 1.59	116.77 ± 0.88
Co^2+^	135.66 ± 1.43	113.80 ± 0.20
Mn^2+^	154.52 ± 0.60	119.54 ± 2.23
Mg^2+^	76.85 ± 1.83	121.62 ± 2.07
K^+^	64.67 ± 1.77	113.51 ± 3.79

**Table 3 marinedrugs-21-00528-t003:** Proportion of products for dextran hydrolysis.

Reaction Time (h)	Hydrolysates (%)			
	Maltoheptanose	Maltohexanose	Maltopentose	Maltotriose
0.5	46.35	29.34	3.99	20.32
1	46.78	29.39	3.83	19.99
2	44.23	30.78	3.97	21.03
3	45.87	30.06	3.80	20.27
4	46.82	29. 20	3.73	20.25

**Table 4 marinedrugs-21-00528-t004:** Water and oil absorption rates of corn porous starch.

Time (h)	Water Absorption (%)	Oil Absorption (%)
0	75.38 ± 2.29	55.63 ± 2.48
6	100.02 ± 0.80	92.94 ± 1.41
12	82.71 ± 2.91	100.00 ± 1.60

## Data Availability

The datasets generated during and/or analyzed during the current study are available from the corresponding author on reasonable request.

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
