# Peer review of "Dextranase Production Using Marine Microbacterium sp. XD05 and Its Application"

_marinedrugs, 2023, doi:10.3390/md21100528_

Round 1
Reviewer 1 Report
This article describes the production and characterization of a marine bacterial dextranase. Overall, the manuscript would be of interest to the readers of this journal. The manuscript can be improved by responding to the following comments and questions:
1. Reference 6 describes the purification and characterization of a dextranase from a soil fungus. It's not clear why this paper was referenced. What specific properties differ between this fungal dextranase and a marine bacteria dextranase?
2. On page 3, Section 2.3.2 The text refers to Figure 4a-e not Figure 3a-e.
3. Figure 4c depicts 6-10% NaCl but line 94 on page 3 states 1%-7%. Which is correct?
4. Page 4 line 100-101, The term production is used several times throughout this 2.4 section but it appears the authors are making the assumption that activity is directly correlated with enzyme production. Higher activity equals high enzyme production. Is there any quantification of enzyme to support this? Was SDS page analysis carried out?
5. Please review Section 2.5.1. The effect of temperature is shown in Figure 7c not Figure 7a. Also, further details should be provided to explain the difference between Figure 7a and Figure 7b. It's not clear why optimum pH is 7.0 in Figure 7a and pH 8.0 in Figure 7b. According to the 4.2.9. section, the enzymes were pre-heated in the different buffers for 4 hours then assayed for activity in Figure 7b.
6.Table 2. NH4+ needs to corrected. 4 should be subscript.
7. line 146, page 6. maltoheptanose and maltohexanose are mispelled.
8. Figure 8a. Is this the chromatogram of the sugar standards?
9. Figure 8b. Maltopentose is listed as a product but not labelled on the Figure. How was the 3% determined? The percentage of peaks looks fairly consistent over time. Did the intensity increase or decrease with time?
10. Section 5 line 339. The authors refer to a purified enzyme, however there are no methods provided in the material and methods section or SDS-page analysis of the crude or purified enzyme. Please provide more details on the procedure if they are available.
English language is satisfactory. Only detected a few typos.
Author Response
Dear Editors and Reviewers,
We really appreciate your kind consideration in giving us an opportunity to revise our manuscript entitled “Dextranase production by marine Microbacterium sp. XD05 and its application” (Ref. No.: marinedrugs-2625150) In the revision, we have revised our manuscript carefully. We appreciate the editors and reviewers for the specific comments and suggestions on our manuscript. We have tried our best to revise the manuscript according to the comments. Revised portions are marked in red in the manuscript. The main corrections in the paper and the responses to the comments are as follows.
Response to the reviewer’s comments:
Point 1: Reference 6 describes the purification and characterization of a dextranase from a soil fungus. It's not clear why this paper was referenced. What specific properties differ between this fungal dextranase and a marine bacteria dextranase?
Response: Thank you very much for the comment. Reference 6 details the purification and characterization of a dextranase from a soil fungus, which is classified as a terrestrial secreted dextranase. It was cited as an introductory comparison point between marine-derived dextranases and their terrestrial counterparts.
Point 2: On page 3, Section 2.3.2 The text refers to Figure 4a-e not Figure 3a-e.
Response: Thank you very much for the comment. It has been revised in the manuscript and marked in red.
Point 3: Figure 4c depicts 6-10% NaCl but line 94 on page 3 states 1%-7%. Which is correct?
Response: Thank you very much for the comment. We apologize for the mistake. It was a data entry error. It is supposed to range from 2% to 6%, with 3% being the optimum. It has been revised in the manuscript and marked in red.
Point 4: Page 4 line 100-101, The term production is used several times throughout this 2.4 section but it appears the authors are making the assumption that activity is directly correlated with enzyme production. Higher activity equals high enzyme production. Is there any quantification of enzyme to support this? Was SDS page analysis carried out?
Response: Thank you very much for the comments. Higher activity of a dextranase enzyme means it is more efficient at breaking down dextran molecules. As only single factor was changed in each experiment, so we assumed the factors were affect quantity of the enzyme production. The primary focus of section 2.4 is the optimization of dextranase production. Yes, the activity of dextranase might be enhanced by some material of the factors. We conducted multiple SDS-PAGE analysis for its molecular weight and concentration. However, we could not accurately determine it in the gel yet.
Point 5: Please review Section 2.5.1. The effect of temperature is shown in Figure 7c not Figure 7a. Also, further details should be provided to explain the difference between Figure 7a and Figure 7b. It's not clear why optimum pH is 7.0 in Figure 7a and pH 8.0 in Figure 7b. According to the 4.2.9. section, the enzymes were pre-heated in the different buffers for 4 hours then assayed for activity in Figure 7b.
Response: Thank you very much for the comments. It has been revised in the manuscript and marked in red. To evaluate thermal stability of our dextranase, the enzyme was exposed to different temperatures (40°C, 45°C, 50°C) for 5 hours without adding its substrate, and the relative residual activity was determined to measure how well the enzyme maintained its functionality under these temperature conditions. For pH stability evaluation, the enzyme was placed in buffers with pH values ranging from 4.0 to 9.0 at a constant temperature of 40°C for 4 hours. The enzyme's viability was then measured using dextran 20000 as a substrate under standard conditions.
Point 6: Table 2. NH4+ needs to corrected. 4 should be subscript.
Response: Thank you very much for the comment. It has been revised in the manuscript and marked in red.
Point 7: line 146, page 6. maltoheptanose and maltohexanose are mispelled.
Response: Thank you very much for the comment. It has been revised in the manuscript and marked in red.
Point 8: Figure 8a. Is this the chromatogram of the sugar standards?
Response: Thank you very much for your comment. Yes, it is the chromatogram of the sugar standards. And the Figure 8b is the chromatogram represents the proportion of products resulting from the hydrolyzed dextran. It has been revised in the manuscript and marked in red.
Point 9: Maltopentose is listed as a product but not labelled on the Figure. How was the 3% determined? The percentage of peaks looks fairly consistent over time. Did the intensity increase or decrease with time.
Response: Thank you very much for the comment. Based on our High-Performance Liquid Chromatography (HPLC) results, we have compiled a table that quantifies the peak areas representing the extent of dextran hydrolysis catalyzed by the dextranase. These products were generated at various time intervals during the enzymatic breakdown of dextran T2000. Notably, the proportion of hydrolyzed dextran attributed to maltopentose after 30 minutes was found to be 3.99%, which is significantly lower compared to the other products. The differences in hydrolysis levels account for the significantly reduced peak intensity of maltopentose compared to the other hydrolysis products to the point that the maltopentose peak is hardly noticeable on the figure.
Point 10: Section 5 line 339. The authors refer to a purified enzyme, however there are no methods provided in the material and methods section or SDS-page analysis of the crude or purified enzyme. Please provide more details on the procedure if they are available.
Response: Thank you very much for the comments. We have added additional details about the purification process in Section 4.2.4 of the manuscript and marked in red.
We would like to express our great appreciation to you for the comments and recognition on our manuscript again.
Sincerely Yours,
Hind Boualis
Jiangsu Key Laboratory of Marine Bioresources and Environment/Jiangsu Key Laboratory of Marine Biotechnology, Jiangsu Ocean University
59 Cangwu Road, Lianyungang, 222005, China.

Reviewer 2 Report
The authors report the isolation and identification of a marine bacterium from the Qingdao Sea that produced a dextranase. The optimal culture conditions of this isolate, identified as Microbacterium sp., was determined with yeast extract and sodium nitrate as the best carbon and nitrogen sources, respectively. The optimal growth temperature was 35°C, pH 6, and 7% NaCl. In addition to culture conditions, the optimum conditions for dextranase production were also determined. Sucrose and ammonium sulfate were the optimal carbon and nitrogen sources, respectively, with optimal temperature of 40°C and pH 7.5. Finally, the optimal conditions for dextranase activity were determined to be 40°C and pH 6-8. The enzyme maintained almost 80% of its activity following storage at 40°C and pH 8 for five hours.
Specific Comments:
1. Figure 3: The legend could contain additional details. For example, was the OD600 measured after 48 hours?
2. Figure 4: Graph D is not needed in manuscript – it could be moved to supplementary. What were the conditions for the growth curve in graph E? The optimal conditions determined in other graphs?
3. Figure 5: What conditions for the 100% relative activity? Is this after 48 hours growth?
4. Did the authors look at metal chelators such as EDTA or EGTA?
None
Author Response
Dear Editor and Reviewer,
We really appreciate your kind consideration in giving us an opportunity to revise our manuscript entitled “Dextranase production by marine Microbacterium sp. XD05 and its application” (marinedrugs-2625150). In the revision, we have revised our manuscript carefully. We appreciate the editors and reviewers for the specific comments and suggestions on our manuscript. We have tried our best to revise our manuscript according to the comments. Revised portions are marked in red in the manuscript. The main corrections in the paper and the responds to the comments are as following.
Responses to the reviewer’s comments:
Point 1: Figure 3: The legend could contain additional details. For example, was the OD600 measured after 48 hours?
Response: Thank you very much for the comment. The OD600 was measured after 24 hours of inoculation. It has been revised in the manuscript and marked in red.
Point 2: Figure 4: Graph D is not needed in the manuscript – it could be moved to supplementary. What were the conditions for the growth curve in graph E? The optimal conditions determined in other graphs?
Response: Thank you very much for the comments. Graph D has been moved from the manuscript. Regarding the growth conditions for strain XD05, the adapted medium based on the results obtained from the previous experiments (optimum growth pH 6 and 3% NaCl), was inoculated with a 2.5% strain inoculation and maintained at 35°C. OD600 was measured every two hours to analyze the growth curve.
Point 3: Figure 5: What conditions for the 100% relative activity? Is this after 48 hours growth?
Response: Thank you very much for the comments. The fermentation mediums were replaced by carbon and nitrogen sources. After being cultured at 35 °C, 180 rpm for 48 h, the supernatant of the fermentation broth was taken to determine the activity of dextranase. Then, the enzyme activity percentages for each source were calculated, compared, and identified, with the sources showing the highest enzyme activity being designated as 100%.
Point 4: Did the authors look at metal chelators such as EDTA or EGTA?
Response: Thank you very much for your suggestion; we appreciate it. We will certainly look into it and incorporate it into our experimental process. EDTA and EGTA serve as reagents to maintain metal ion-free conditions in chemical reactions and assays. Furthermore, EDTA functions as a preservative in food and cosmetic products to prevent degradation caused by metal ions. In our study of dextranase, we conducted analyses in the presence of varying concentrations of metal ions commonly found as contaminants in industrial settings, known to inhibit enzymatic activity. Our results indicate that XD05 dextranase retains enzymatic activity even in the presence of trace amounts of these metal ions, which holds promise for its industrial applications.
We would like to express our great appreciation to you for the comments and recognition on our manuscript again.
Sincerely Yours,
Hind Boualis
Jiangsu Key Laboratory of Marine Bioresources and Environment/Jiangsu Key Laboratory of Marine Biotechnology, Jiangsu Ocean University
59 Cangwu Road, Lianyungang, 222005, China.

Round 2
Reviewer 1 Report
The authors have addressed most of the comments. There is one error on page 4 line 96: Figure 4e should be Figure 4d.
Some minor spelling mistakes to revise.